

# The impact of QBO disruptions on diurnal tides over the low- and mid-latitude MLT region observed by a meteor radar chain

Jianyuan Wang[1,2,3,4], Na Li[1,2], Wen Yi[3,4,*], Xianghui Xue[3,4,5,6,*], Iain M. Reid[7,8], Jianfei Wu[3,4], Hailun Ye[3,4], Jian Li[3,4], Zonghua Ding[1,2], Jinsong Chen[1,2], Guozhu Li[9], Yaoyu Tian[1], Boyuan Chang[1], Jiajing Wu[1] and Lei Zhao[1,2]

[1]National Key Laboratory of Electromagnetic Environment, China Research Institute of Radiowave Propagation, Qingdao 266107, China
[2]Kunming Electro-magnetic Environment Observation and Research Station, Qujing 655500, China
[3]CAS Key Laboratory of Geospace Environment, Department of Geophysics and Planetary Sciences, University of Science and Technology of China, Hefei, China
[4]CAS Center for Excellence in Comparative Planetology, Anhui Mengcheng Geophysics National Observation and Research Station, University of Science and Technology of China, Hefei, China
[5]Hefei National Laboratory, University of Science and Technology of China, Hefei, China
[6]Collaborate Innovation Center of Astronautical Science and Technology, Harbin 150001, China
[7]ATRAD Pty Ltd., Adelaide, SA 5032, Australia
[8]School of Physical Sciences, University of Adelaide, Adelaide, SA 5005, Australia
[9]Key Laboratory of Earth and Planetary Physics, Institute of Geology and Geophysics, Chinese Academy of Sciences, Beijing, China

*Correspondence to*: Xianghui Xue (xuexh@ustc.edu.cn) and Wen Yi (yiwen@ustc.edu.cn)

**Abstract.** A quasi-biennial oscillation (QBO) disruption is a very rare phenomenon in which QBO westward wind is temporarily interrupted by the occurrence of a band of westward wind in the tropical stratosphere. This phenomenon is important as it could greatly affect the global atmospheric circulation, especially in the mesosphere. Past observational and modelling studies have shown the QBO-varying mesospheric diurnal tide, but the mechanism is still not fully understood. In this study, we report the strong response of mesospheric diurnal tides to the two QBO disruptions that occurred in 2015/16 and 2019/20 and its possible mechanisms. The diurnal tidal winds are observed by a meteor radar chain, consisting of meteor radars located at Kunming (25.6°N, 103.8°E), Wuhan (30.5°N, 114.2°E), Mengcheng (33.4°N, 116.5°E), Beijing (40.3°N, 116.2°E) and Mohe (53.5°N, 122.3°E) in China. These observations provide clear evidence that mesospheric diurnal tides are unusually weakened (by ~-6 m/s) during these QBO disruptions, over Kunming, Wuhan, Mengcheng, and Beijing. By using the Specific Dynamics version of the Whole Atmosphere Community Climate Model with thermosphere and ionosphere extension (SD-WACCM-X) and the European Centre for Medium-Range Weather Forecasts (ECMWF) Reanalysis v5 (ERA5) dataset, the analysis indicates that the QBO wind affects mid-latitude mesospheric diurnal tides by modulating both the solar radiative absorption by subtropical stratospheric ozone (~5 to 0.5 hPa) and the tidal-gravity wave interaction in the mesosphere (~60 to 100 km). Thus, these unexpected QBO disruptions provide an opportunity to better understand the coupling between climate change and middle atmospheric dynamics.





**Short Summary.** The mesospheric diurnal tides over low- and mid-latitude region are suppressed during the QBO disruptions and QBO westward wind phase, and are enhanced during QBO eastward wind phase, observed by a meteor radar chain. By using SD-WACCM-X simulations and ERA5 reanalysis, it is found that the stratospheric QBO winds affect the mesospheric diurnal tides by modulating the subtropical ozone variability in the upper stratosphere and the interaction between tides and gravity waves in the mesosphere.

## 1 Introduction

Atmospheric tides are global-scale atmospheric oscillations with periods that are harmonics of a solar day (Chapman and Lindzen, 1970). The diurnal tides (24-hour period) are dominant modes that have been extensively studied. Numerous studies have reported interannual variabilities in diurnal tides (Pancheva et al., 2020; Davis et al., 2013; Hagan et al., 1999; Laskar et al., 2016; Lieberman et al., 2004; Lieberman, 1997; He et al., 2024). The interannual variability in these tides is mainly
attributed to the 11-year solar cycle (Sun et al., 2022), the El Niño Southern Oscillation (ENSO; Cen et al., 2022; Lieberman et al., 2007) and the quasi-biennial oscillation (QBO; Davis et al., 2013; Hagan et al., 1999; Laskar et al., 2016; Salinas et al., 2023).

The QBO in the tropical stratosphere is the dominant mode of interannual variability in the zonal mean zonal wind in the pressure range of 5-100 hPa; it consists of the descent of alternating eastward and westward winds with a period of
approximately 20-30 months (Ebdon, 1960; Reed et al., 1961; Andrews et al., 1987). Theory states that the tropical stratospheric QBO is driven by upward propagating equatorial planetary waves and GWs via momentum deposition (Lindzen and Holton, 1968; Plumb and McEwan, 1978; Baldwin et al., 2001). The QBO can strongly modulate stratospheric dynamic processes such as the ozone transport from tropical to high latitude regions (Hampson and Haynes, 2006; Holton and Tan, 1980); in addition to the stratosphere, the QBO modulates deep convection in the troposphere (Collimore et al., 2003), the
Madden-Julian oscillation (Zhang and Zhang, 2018), as well as the propagation of atmospheric tides (Davis et al., 2013; Hagan et al., 1999; Laskar et al., 2016), planetary waves (Andrews et al., 1987), and GWs (Geller et al., 2016). By modulating the atmospheric waves that propagate vertically from the troposphere to the mesosphere, the QBO signature can reach higher altitudes and play a significant role in middle atmospheric dynamics. For example, the QBO is clearly evident in low latitude mesospheric winds (e.g., Vincent et al., 1998) and OH (~85 km) and OI (~96 km) nightglow (e.g., Reid et al., 2014), and the
impact of the QBO on atmospheric tides is an important dynamic process.

In the past, several studies have reported the QBO variabilities of tides in the mesosphere and lower thermosphere (MLT) region. Vincent et al. (1998) reported a clear QBO-like variability in diurnal tides over Adelaide (35°S, 138°E) observed by long-term MF radar observations. By using the numerical spectral model (NSM), Mayr and Mengel (2005) suggested that the interannual variability in mesospheric diurnal tides is generated by the QBO possibly due to the momentum deposition of GWs.
Using meteor radar-based winds over Andenes (69°N, 16°E) and Juliusruh (54°N, 13°E), Laskar et al. (2016) reported that mesospheric semidiurnal tides are enhanced during the QBO eastward phase (QBOE) and suppressed during the QBO



westward phase (QBOW). These authors suggested that the filtering effect of the QBO zonal wind on planetary waves and the interaction between tides and planetary waves imprint the QBO signature on mesospheric tides. Pramitha, et al. (2021) suggested that the QBO variation in mesospheric diurnal tides at low latitude is associated with ozone variability at the QBO scale. There are three primary theories to explain the observed QBO signature in mesospheric diurnal tides. These are based on: GW momentum deposition, the filtering effect of zonal mean flow and stratospheric ozone heating. However, existing observational evidence cannot conclusively support these conjectures, so the mechanism of this process is still unclear.

During the 2015/16 winter, a very rare phenomenon occurred in the tropical stratosphere in the form of the temporarily interrupted QBO eastward wind by a band of developing westward wind; this phenomenon is called the QBO disruption (Osprey et al., 2016; Newman et al., 2016; Coy et al., 2017; Kang et al., 2020; He et al., 2022). Several studies suggested that this phenomenon is forced by anomalously enhanced westward equatorial Rossby waves (Osprey et al., 2016; Newman et al., 2016). Barton and McCormack (2017) suggested that an extreme El Niño event induces the QBO disruption by weakening the lower stratospheric subtropical westward jet. Pramitha et al. (2021) first reported a clearly disrupted QBO signature in low-latitude mesospheric diurnal tides observed by Tirupati (13.63°N, 79.4°E) meteor radar. However, the extreme El Niño event during November 2015-January 2016 also suppressed the mesospheric diurnal tides (e.g., Cen et al., 2022); therefore, the impact of the disrupted QBO on mesospheric diurnal tides is still poorly understood.

Unexpectedly, the QBO was interrupted again in the 2019/20 winter with lower El Niño anomalies (Li et al., 2023; Kang and Chun, 2021; Kang et al., 2022), providing a valuable opportunity to further understand the connection between stratospheric QBO disruptions and mesospheric diurnal tides. In addition, the stratospheric QBO can strongly modulate global circulation from the tropics to the poles, but there are few reports on how QBO disruptions affect the mid-latitude mesospheric dynamics. In this regard, this study reports the impacts of the 2015/16 and 2019/20 QBO disruptions on mid-latitude mesospheric diurnal tides observed by a meteor radar chain, consisting of five meteor radars located at Kunming (25.6°N, 103.8°E), Wuhan (30.5°N, 114.2°E), Beijing (40.3°N, 116.2°E) and Mohe (53.5°N, 122.3°E). The ERA5 reanalysis dataset and the Specific Dynamics version of the Whole Atmosphere Community Climate Model with thermosphere and ionosphere extension (SD-WACCM-X) simulations are also used to determine the possible mechanism responsible for the connection between tropical stratospheric QBO disruptions and mid-latitude mesospheric diurnal tides. Section 2 provides the data and methods employed in this study; section 3 presents the results; and the discussion and conclusions are provided in sections 4 and 5, respectively.

## 2 Data and Methodology

### 2.1 Meteor radars

Meteor radars are used to calculate the horizontal wind over Mohe, Beijing, Mengcheng, Wuhan and Kunming. Table 1 summarizes the basic system parameters, geographic coordinates and observational time-periods for the meteor radars used in this study. Figure 1 presents the geographic locations of these meteor radars and the variability in the QBO magnitude with latitude. These meteor radars are ATRAD meteor detection radar (MDR) series (Li et al., 2018; Holdsworth et al., 2004; Yi et



al., 2019; Zhou et al., 2022). The meteor radars located at Mohe, Beijing and Wuhan belong to the Institute of Geology and

Geophysics, Chinese Academy of Sciences (IGGCAS), which is part of the Chinese Meridian Project and STERN (the Solar-Terrestrial Environment Research Network). The Mengcheng meteor radar is run by University of Science and Technology of China. The Kunming meteor radar is run by Kunming Electro-magnetic Environment Observation and Research Station.

**Table 1: Geographic locations, operation frequencies and observational time periods of the meteor radars used in this study.**

| Meteor radar | Geographic locations | Frequency | Data used in this study |
|---|---|---|---|
| Mohe | 53.5° N, 122.3° E | 38.9 MHz | 2011/8/1-2023/1/1 |
| Beijing | 40.3° N, 116.2° E | 38.9 MHz | 2008/12/4-2023/1/1 |
| Mengcheng | 33.4° N, 116.5° E | 38.9 MHz | 2014/3/15-2023/1/1 |
| Wuhan | 30.5° N, 114.2° E | 38.9 MHz | 2010/9/22-2023/1/1 |
| Kunming | 25.6° N, 103.8° E | 37.5 MHz | 2008/7/29-2022/9/13 |

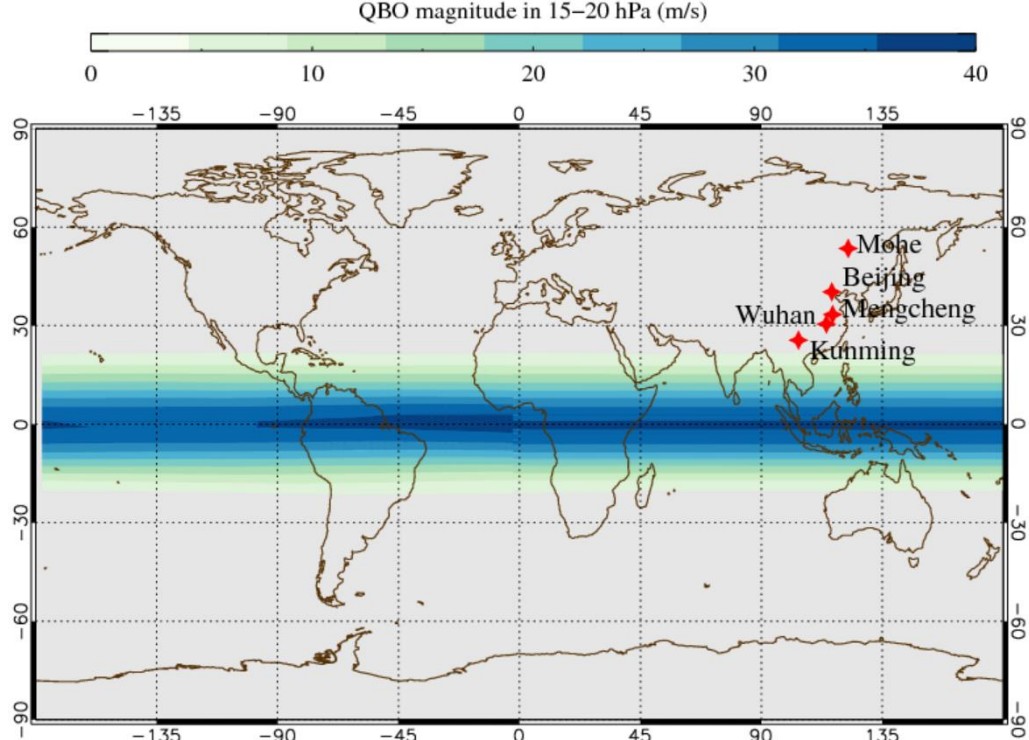


**Figure 1: The QBO magnitude (m/s) averaged in the pressure level range of 15-20 hPa during 2000-2014 according to the ERA5 reanalysis. Before calculating the QBO magnitude, the monthly mean zonal wind is derived by removing seasonal variations and the semiannual oscillation (SAO). The red stars mark the locations of the five radars used in this study.**

Meteor radar operation and analysis for these radars is described by Holdsworth, et al. (2004) and the interested reader is

directed there. All five radars are used to measure horizontal winds and sample in the altitude region from 70 to 110 km. The





temporal and altitudinal resolutions are 1 hour and 2 km, respectively. Because of missing horizontal wind data in the altitude ranges of 70-76 km and 100-110 km, only the observed horizontal winds in the altitude range of 78-98 km are analyzed in this study.

## 2.2 SD-WACCM-X

SD-WACCM-X is a comprehensive numerical model based on the Community Earth System Model version 1 (CESM1) framework (Hurrell et al., 2006), and is designed to investigate the coupling among chemistry, radiation, and dynamics and their impact on the Earth's climate system (Neale et al., 2013). The SD-WACCM-X version 2.1 simulation from the surface up to ~50 km is nudged from Modern-Era Retrospective analysis for Research and Applications, Version 2 (MERRA-2) data. In this study, the monthly SD-WACCM-X simulation is used to obtain the mesospheric wind diurnal tides, heating sources of

these tides and GW drag on zonal wind; these heating sources in the SD-WACCM-X are exported as temperature tendencies due to the solar heating rate, solar radiative absorption by water vapor and ozone (mW/kg); the GW drag on the zonal wind in the SD-WACCM-X is exported as zonal wind tendencies due to total GW drag (m/s/day); and the tropical stratospheric zonal mean zonal wind (m/s) in the SD-WACCM-X output is also used to present the QBO signature in tropical stratospheric GWs.

## 2.3 ERA5 reanalysis

ERA5 is the fifth-generation reanalysis dataset from the ECMWF. It provides several improvements compared to ERA-I, as detailed by Hersbach and Dee (2016). The analysis is produced at a 1-hourly time step using a significantly more advanced 4D-var assimilation scheme. Its horizontal resolution is approximately 31 km, and atmospheric variables are calculated at 137 pressure levels (Hersbach et al., 2020). The data for the 1979–2022 period is released in 2023. In this study, the monthly ozone concentration (mg/kg) in model levels is used to analyze the possible excitation sources of mesospheric diurnal tides.

**2.4 Approach for tidal decomposition**

The series of hourly zonal and meridional winds are performed by least squares fitting in a 3-day sliding window with a 1-day time step to decompose the amplitudes and phases of various tidal components, including diurnal, semidiurnal and terdiurnal tides. The screening conditions for the fitting are as follows: if the valid data rate within the window is less than half the total or span less than two-thirds of their phase, the data within this window cannot be used for subsequent calculation. Otherwise,

the 3-day wind data remains eligible for analysis. Then, a least square fitting method, as described by Baumgarten and Stober (2019), is performed on the hourly wind data for each 3-day window throughout the series to decompose the diurnal tides as follows.

$$u(z, t) = u_0(z) + \sum_{n=1}^{3} a_n(z) \sin\left(\frac{2\pi n}{T} t\right) + b_n(z) \cos\left(\frac{2\pi n}{T} t\right), \tag{1}$$

where $u$ represents either the zonal or meridional wind; $z$ represents altitude; $t$ represents time; $n$ represents temporal

wavenumber; $T$ equals 24 hours; and $u_0$ represents mean zonal or mean meridional wind.





## 3 Results

### 3.1 Meteor radars observations

Figure 2a presents the zonal wind in the pressure level range of 100-10 hPa observed by the Singapore radiosonde (1°N, 104°E), which reveals the normal pattern of the QBO with alternately descending westward and eastward wind with a period of
approximately 20-30 months. This characteristic QBO zonal wind pattern is disrupted in 2015-2016 and 2019-2020, as highlighted by dotted lines in Figure 2a. During the two QBO disruptions, the descending eastward wind is interrupted by a localized westward wind near the pressure level of 40 hPa, resulting in split equatorial westerly jets. In this study, we focus on the strong impacts of the recent QBO disruptions on mid-latitude MLT diurnal tides.

Figures 1b-1f present the meridional diurnal tidal amplitude perturbations over Kunming, Wuhan, Mengcheng, Beijing and
Mohe, respectively. To show the connection between diurnal tides and the stratospheric QBO, diurnal tidal perturbations are derived by removing seasonal variations and the 11-year solar cycle variations. Diurnal tidal perturbations over Kunming, Wuhan, Mengcheng and Beijing are very similar to the variability in the QBO zonal wind, while the diurnal tides over Mohe do not exhibit a clear QBO signature. Diurnal tidal perturbations are enhanced during QBOE and those are suppressed during QBOW. However, during 2015-2017 and 2019-2021, diurnal tides in the MLT region are unusually weakened when the
eastward wind is interrupted by a localized westward wind near 40 hPa. The tidal responses to the QBO disruptions were remarkably similar over Beijing, Mengcheng and Wuhan (~30-40°N). In contrast, the diurnal tides observed by Kunming meteor radar (~25°N) primarily respond to the QBO disruption above 80 km. This suggests that the impact of the QBO disruptions on mesospheric diurnal tides might differ between the subtropical region and in the mid-latitude region.





**Figure 2: (a) The QBO zonal wind observed by the Singapore radiosonde for the 100-10 hPa pressure levels (~15-30 km). The red curve indicates the Niño 3.4 index. (b-f) The meridional diurnal tidal amplitude perturbations observed from meteor radars over (b) Kunming, (c) Wuhan, (d) Mengcheng, (e) Beijing and (f) Mohe in the altitude range from 78 to 98 km during 2008-2023. These tidal perturbations are derived by removing the seasonal variations and 11-year solar cycle variations. Note that the color bar values are different. The dashed lines represent QBOE and QBOW. The red and blue solid arrows denote the QBOE and QBOW, respectively. The blue hollow arrows denote the two QBO disruptions in 2015/16 and 2019/20 winter.**

In the past, numerous studies have reported the QBO signature in mesospheric diurnal tides (Davis et al., 2013; Hagan et al., 1999; Laskar et al., 2016; Vincent et al., 1998; Mayr and Mengel, 2005; Ern et al., 2014; Ern et al., 2023; Salinas et al., 2023). However, the features and mechanism of mesospheric tides during QBO disruptions are still poorly understood. The ENSO, as the dominant interannual variation in the tropical troposphere (Yulaeva and Wallace, 1994), can also affect mesospheric diurnal tides (Cen et al., 2022; Lieberman et al., 2007). Cen et al. (2022) suggested that El Niño events suppress mesospheric diurnal tides in the boreal winter, including the 2015/16 winter. Figure 2e presents the Niño 3.4 index (red curve) during 2008-





2023. The Niño 3.4 index reached a temperature anomaly of 2.57 K during the 2015/16 winter, while it reached 0.74 K during the 2019/20 winter. As shown in Figures 2a-2c, the negative mesospheric tidal perturbations during the 2015/16 winter exhibit weaker intensity compared to those during the 2019/20 winter. This pattern contrasts with the Niño 3.4 index suggesting that
the negative tidal perturbations observed during QBO disruptions are not linked to the ENSO. Therefore, the negative tidal perturbations observed during 2015/16 winter and 2019/20 winter are likely associated with the QBO disruptions.

## 3.2 ERA5 ozone variability

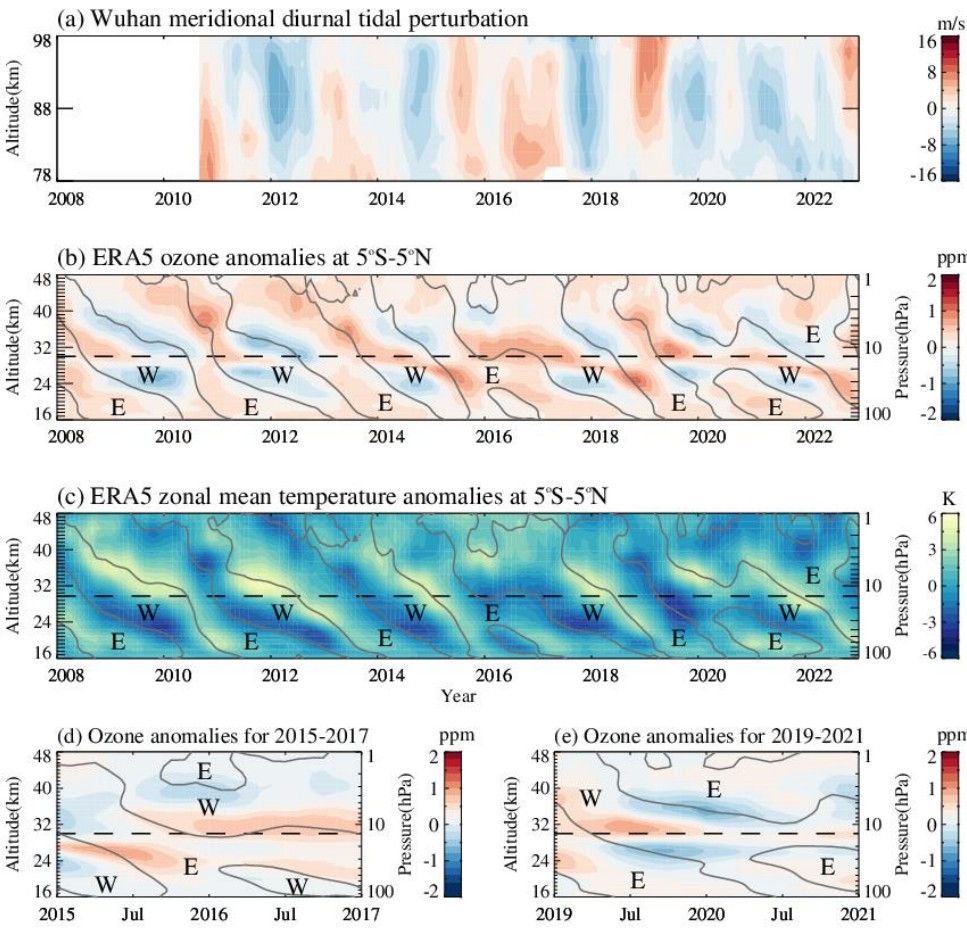

**Figure 3: (a) The meridional diurnal tidal amplitude perturbations observed from the meteor radar over Wuhan in the altitude**
**range from 78 to 98 km during 2008-2023. (b) The monthly mean anomalies of ozone mixing ratio from ERA5 reanalysis (contour fill plots; ppm: $10^{-6}$ kg/kg) averaged between 5°S and 5°N in the altitude range of 20-60 km. The grey solid lines indicate the zero-wind lines of monthly averaged zonal mean zonal wind (E: eastward wind; W: westward wind). (c) As in (b), but for zonal mean temperature. The anomalies are derived by removing the seasonal variations and 11-year solar cycle variations. The dashed lines denote the altitude of 30 km.**



Considering that QBOs dominate the stratospheric dynamic in the equatorial region and the major tidal heating source in the stratosphere is the solar radiative absorption by ozone molecules, the possible QBO impact on ozone concentrations may cause a QBO signature in stratospheric solar radiative heating. In this study, the ERA5 ozone concentration is used to explore the QBO impact on ozone variability and the relationship between ozone variability and observed diurnal tides. Figure 3 presents the anomalies in the ozone concentration and QBO in the tropical stratosphere (5°S-5°N). The ozone variability is associated

with the stratospheric QBO wind (grey solid lines in Figure 3b) and the mid-latitude mesospheric diurnal tidal perturbations (Figure 3a).

As shown in Figure 3b, the tropical stratospheric ozone is strongly modulated by a QBO-related variability, primarily within the altitude range of 20-50 km. The ozone concentration reaches the minimum value within the altitude range of ~22-28 km and ~30-40 km respectively as the zonal mean zonal wind turns to westward within the altitude range of ~20-35 km (as shown

in Figure 3d). When the zonal mean zonal wind at altitudes ranging from ~20-35 km is dominated by eastward wind, the ozone concentration reaches the maximum value within the altitude of ~22-28 km and ~32-45 km respectively. In the lower stratosphere (~22-28 km), positive (negative) ozone anomalies propagate downward in the similar form of zonal mean eastward (westward) wind shears. This behavior aligns with the zonal mean positive (negative) temperature anomalies as depicted in Figure 3c. In the upper stratosphere (above ~30 km), the phase of ozone anomalies is almost opposite to the phase of QBO

winds and temperature. During the QBO disruptions, both layers of ozone anomalies exhibit negative accordingly in 2015/16 and 2019/20 winter as shown in Figures 3d and 3e.

In the tropical lower stratosphere, the ozone variability is primarily determined by transport due to the relatively longer chemical lifetime of the ozone molecule and stronger vertical gradient in this area. In the tropical upper stratosphere, the ozone variability is modulated by both transport and photochemistry process. In the tropical stratosphere, when the QBO wind shear

is eastward (westward), the air becomes warm (cooling) due to thermal wind balance (Figure 3c; Baldwin et al., 2001). In the lower stratosphere, the QBO-related temperature differences induce a downward (upward) meridional circulation (Gray and Chipperfield, 1990), resulting in the increasing (reduced) ozone and nitrous oxide ($N_2O$) concentration (Salawitch et al., 2005; Park et al., 2017). $N_2O$ is the primary sources of the NOx species and NOx is the major sink of ozone in the middle stratosphere (Salawitch et al., 2005). Thus, in the upper stratosphere, the phase of ozone anomalies is opposite to the phase of the QBO-

related temperature due to chemical control by NOx (Park et al., 2017). As a result of alternating descending pattern of QBO winds, the phase of ozone anomalies in the upper stratosphere is approximately coherent with QBO wind at ~20 hPa. Besides, the solar radiative absorption by the ozone molecule in the upper stratosphere is also the major heating sources of the diurnal tides (Hagan et al., 1999; Vichare and Rajaram, 2013) and the ozone variability in the upper stratosphere is in a coherent manner with the mesospheric diurnal tides. Thus, the ozone variability in the upper stratosphere is primarily focused in this

study.



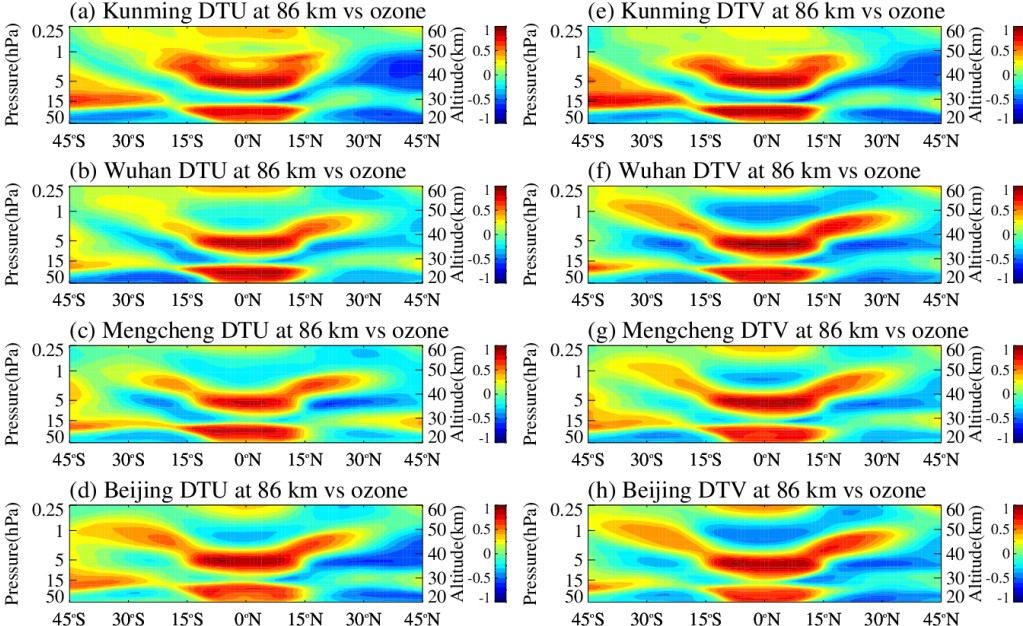

**Figure 4: (a-d) Correlations between the anomalies in the ozone concentration in 45°S-45°N in the altitude range of 20-60 km and zonal diurnal tidal amplitudes at 86 km observed by meteor radars over (a) Kunming, (b) Wuhan, (c) Mengcheng, and (d) Beijing, respectively. (e-h) As in (a-d), but for meridional diurnal tides.**

To investigate the connection between QBO-related ozone variability and mesospheric diurnal tides, Figures 4a-4h present the correlation between the ozone variability and diurnal tidal perturbations in both wind components observed by meteor radars. As shown in Figure 4, although the latitudinal differences in these tidal observations are significant, ranging from ~25°N to ~40°N, the region with a strong correlation between ozone and these tides remains relatively unchanged. A portion of the ozone correlated with tides is located near 30 hPa in the tropical lower stratosphere of approximately 10°S-10°N, and another

portion is situated near 5 hPa in the ~10°S-10°N tropical upper stratosphere, extending toward the subtropical region (~15°N-30°N) as the altitude increases up to ~0.5 hPa.

The correlation analysis suggests that the interannual oscillation of ozone in the tropical lower stratosphere and subtropical upper stratosphere are in phase with the interannual oscillation of diurnal tides over low- and mid-latitude MLT region. As shown above, dominant variability in the diurnal tides over low- and mid-latitude MLT region and the ozone concentration in

the tropical lower and upper stratosphere is tied to the QBO. Interestingly, the region of QBO-related ozone variability in the upper stratosphere is not limited in the tropical region, but extends toward the subtropical region (~30°N) with the altitude range increasing from ~30-40 km to ~40-50 km. In addition to the interannual variability, the relationship between the response of the upper stratospheric ozone and mesospheric diurnal tides to the QBO disruptions also deserves to be discussed.



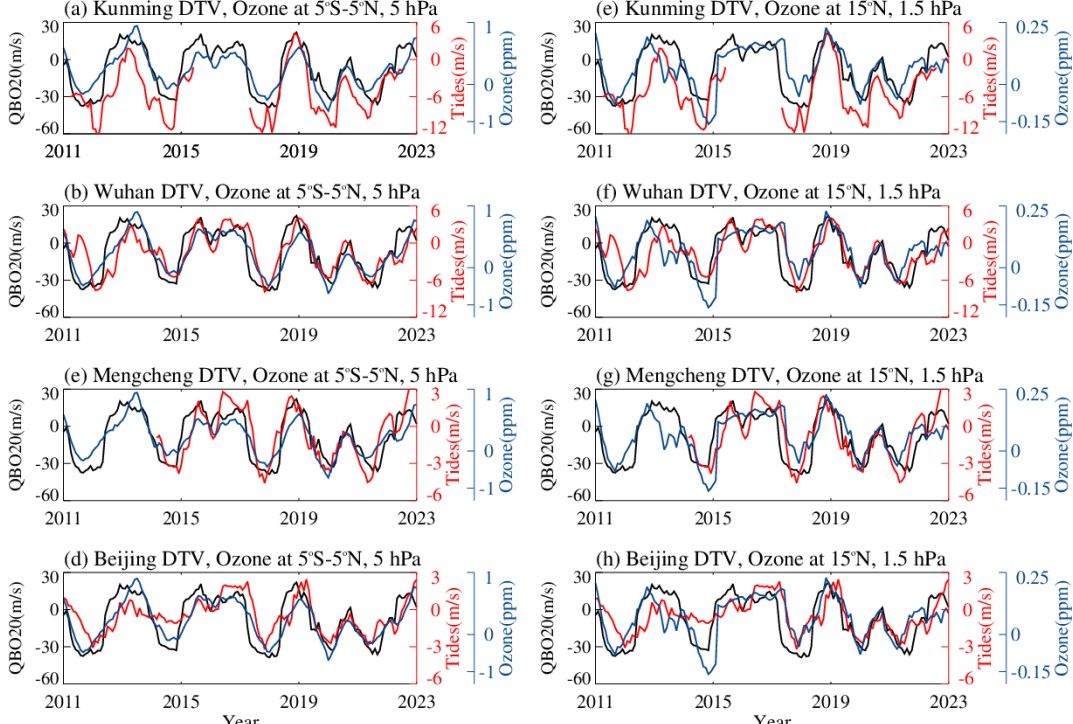

**Figure 5: (a-d) Comparative analysis of the meridional diurnal tides observed by meteor radars (red) over (a) Kunming, (b) Wuhan, (c) Mengcheng and (d) Beijing with the QBO wind observed by Singapore radiosonde at 20 hPa (QBO20, black) and ozone concentration anomalies derived by ERA5 reanalysis (blue) at 5°S-5°N, 5 hPa, respectively. (e-h) As in (a-d), but for ozone anomalies at 15°N, 1.5 hPa. The anomalies in the ozone concentration are derived by removing the seasonal variations and 11-year solar cycle variability.**

To show the QBO signature and QBO disruptions impacts on ozone variability, Figures 5a-5h present the ozone anomalies in the tropical upper stratosphere (5°S-5°N, 5 hPa) and subtropical upper stratosphere (~15°N, 1.5 hPa). As shown in Figures 5a-5d, the meridional diurnal tides at 86 km over Kunming, Wuhan, Mengcheng, and Beijing as well as the ozone variabilities at 5°S-5°N, 5 hPa are all strongly consistent with the QBO wind at 20 hPa. As shown in Figures 5e-5h, the ozone variability at 15°N, and 1.5 hPa also exhibits a significant QBO signature, which is associated with the QBO wind at 20 hPa and with mesospheric diurnal tides. Furthermore, during the 2015/16 and 2019/20 winter, a simultaneous reduction in ozone concentration was observed in both the tropical and subtropical stratosphere. This decreases, along with the attenuation of the mesospheric tides, exhibited a significant alignment with the temporary westward QBO wind pattern.

Of course, statistically significant correlations between two parameters cannot imply their interdependency. Considering that the ozone heating responsible for exciting diurnal tides primarily occurs at altitudes between ~40-50 km (Figure B1; Hagan et al., 1999; Vichare and Rajaram, 2013), ozone concentration variabilities near 30 hPa in the tropical stratosphere are less likely to be the cause of the QBO signature in mesospheric diurnal tides. The connection between QBO-related ozone in ~5-0.5 hPa (~35-50 km) and the QBO signature in mesospheric diurnal tides needs further analysis via using a comprehensive numerical model, such as SD-WACCM-X. The QBO impacts on the tidal excitation are explored in section 4.





## 3.3 SD-WACCM-X simulations

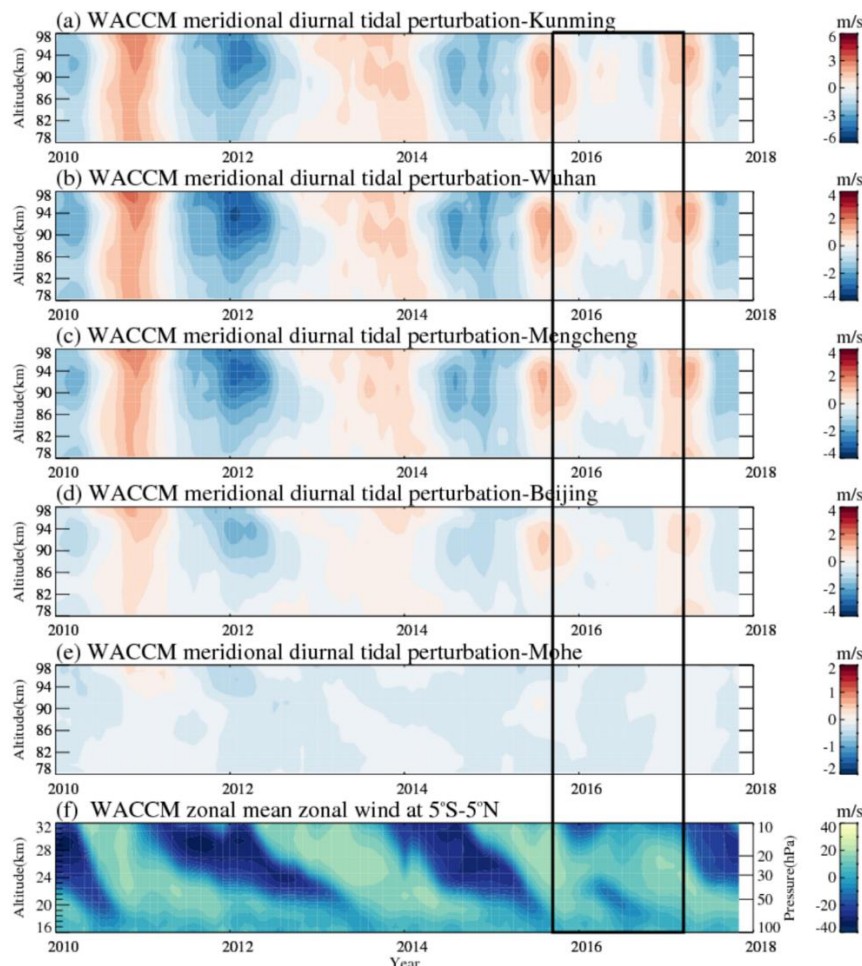


**Figure 6: (a-e) The meridional diurnal tidal amplitude perturbations derived by SD-WACCM-X over (a) Kunming, (b) Wuhan, (c) Mengcheng, (d) Beijing and (e) Mohe in the altitude range from 78 to 98 km during 2010-2018. These tidal perturbations are derived by the same methodology as in Figure 2. Note that the color bar values are different. (f) The zonal mean zonal wind derived by SD-WACCM-X averaged in 5°S-5°N in the pressure levels of 100-10 hPa during 2010-2018. The rectangular box highlights the QBO**
**disruptions in 2015/16 winter.**

To further consider the mechanism of disrupted QBO impacts on tides, the diurnal tides in zonal and meridional components over the locations of these five meteor radars are simulated by SD-WACCM-X in the altitude ranging from 78 km to 98 km. Figure 6 presents the SD-WACCM-X simulations for the meridional diurnal tidal perturbations during 2010-2018, over the locations of Kunming, Wuhan, Mengcheng, Beijing and Mohe. The WACCM tidal perturbations are deseasonalized by the

same methodology to remove the seasonal variations and the 11-year solar cycle variability as on the meteor radar observations. Figure 6f presents the tropical zonal mean zonal wind simulated by WACCM modelling. Compared to the zonal wind observed by Singapore radiosonde (Figure 2a), the WACCM modelling could well simulate the QBO wind and the QBO disruption in 2015/16 winter. As shown in Figure 6, the mid-latitude mesospheric diurnal tides derived by WACCM over Kunming, Wuhan,



Mengcheng, and Beijing are also greatly weakened during the 2015/16 QBO disruption. Similarly, the QBO signature can be
clearly identified in the diurnal tidal perturbations over Kunming, Wuhan, Mengcheng, and Beijing during the simulated time-period that the mesospheric diurnal tides are enhanced during QBOE and suppressed during QBOW.

However, compared to meteor radar observations, the QBO signature in diurnal tides simulated by WACCM is ~3 m/s over
Kunming is weaker than the observations value of ~6 m/s. Also, the time-period during that the diurnal tides are weakened
during the QBO disruption in the WACCM output is longer than that observed by meteor radars. Stober et al. (2021) and Zhou
et al. (2022) have compared the mesospheric tides observed by meteor radars and simulated by WACCM modelling, and
discussed the agreements and deviations of the seasonal variations in mean winds and tides. Their results also suggested that
the meteor radar observed diurnal tidal amplitudes are usually greater than those simulated by WACCM. The tidal difference
between observations and modelling simulations primarily depends on tidal forcing in the troposphere and stratosphere
(Ortland, 2017), propagation and interaction with GWs (Stober et al., 2021). These differences indicate that these processes
are still unclear in the QBO modulation; nevertheless, WACCM simulations could still be used to explore the mechanism of
QBO modulation considering that disrupted QBO signatures are clearly identified in WACCM diurnal tides.

## 4 Discussion

### 4.1 Tidal forcing and propagation

Diurnal tides are primarily excited by solar radiative absorption by water vapor in the troposphere, ozone molecules in the
stratosphere, and oxygen molecules in the thermosphere. According to tidal theory, the heating rates of solar radiative
absorption by water vapor in the upper troposphere and ozone molecules in the stratosphere are responsible for mesospheric
diurnal tidal forcing. Thus, we examine the perturbation of the diurnal solar heating source simulated by the WACCM
modelling in the altitude range of 0-100 km, which potentially contributes to the positive diurnal tidal anomalies in the
stratosphere during QBOE and negative diurnal tidal anomalies during QBOW.





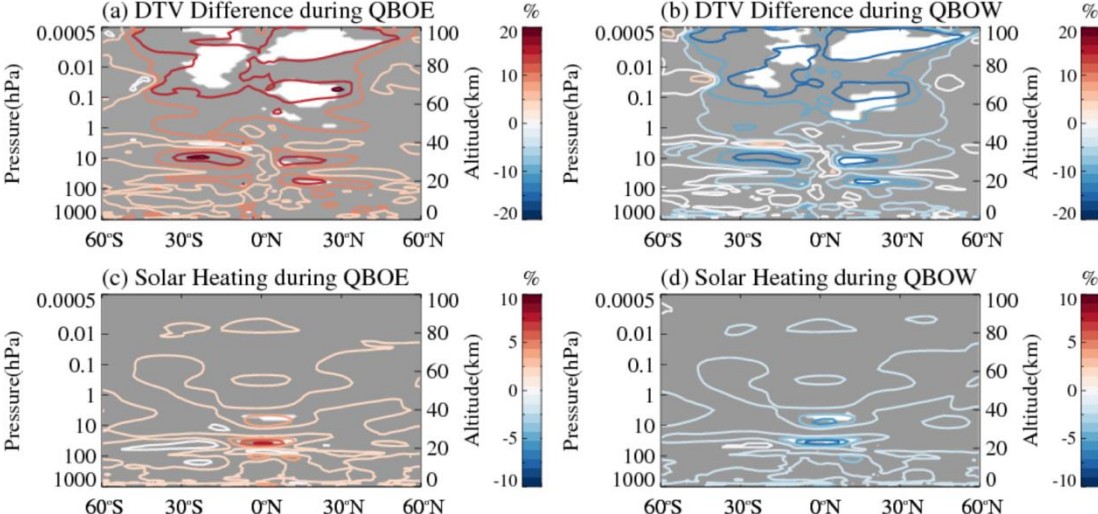

**Figure 7: (a) Anomalous percentage variances of the meridional diurnal tides in the latitudinal range of 60°S-60°N during QBOE. (b) As in (a), but during QBOW. (c) As in (a), but for the solar radiative heating. (d) As in (b), but during QBOW. The solid red lines denote the positive response, while the solid blue lines denote the negative response. The white regions denote 95% significance according to the Monte Carlo test; the grey area indicates where the response is insignificant at the 95% level according to the Monte Carlo test. Note that the color bar values are different.**

Figures 7a and 7b present the anomalous percentage variances of the WACCM meridional diurnal tides averaged in 100-120°E during QBOE and QBOW, respectively. During QBOE, the meridional diurnal tides are greatly enhanced by ~20% in the MLT region (80-100 km) from 45°S to 45°N; the diurnal tides are also significantly positive in the altitude range of ~50-80 km from 5°N to 30°N and in the altitude range of 20-40 km from 5°N to 25°N. During QBOW, the diurnal tides are suppressed by ~-20% in the same region. The altitudinal and latitudinal variabilities in these regions with significantly positive responses suggest that the QBO signature in diurnal tides over the mid-latitude MLT region may propagate from the tropical and/or subtropical stratosphere. The tidal response to the QBO in the WACCM simulation is consistent with that observed by meteor radars, so the WACCM simulation is expected to be used to explore the process of QBO modulations on mesospheric diurnal tides.

As shown in Figures 7c and 7d, the solar radiative heating rates are significantly positive (~10%) in the tropical region (~10°S-10°N) at ~5 hPa and ~30 hPa during QBOE and negative (~-10%) in the same region during QBOW. The region where the solar heating rates strongly respond to QBO extends to mid-latitude with increasing altitude, from ~5 hPa to ~0.1 hPa, although this extended area is not significant at the 95% level. At the altitude of the strong response region at ~5-0.1 hPa, the tidal forcing is primarily solar radiative absorption by ozone molecules. Notably, the QBO-related solar heating variability is approximately in the same region as the QBO-related ozone variability in the upper stratosphere, indicating that the QBO affects the tidal heating source in the stratosphere via modulating the solar radiative absorption by ozone molecules. According to the relationship between ozone variability at ~5-0.5 hPa and QBO-related tidal perturbations in the mid-latitude MLT region, the possible mechanism is as follows:





During QBOE, the dominant eastward zonal wind descends in the tropical stratosphere, and a warm region develops at the
location of maximum vertical wind shear via thermal wind balance (Baldwin et al., 2001). As the tropical lower stratosphere
becomes warmer and the tropical upper stratosphere is cooler at the same time, the infrared cooling to space is relatively
enhanced below ~30 km and is relatively suppressed above ~30 km, resulting in a downward-poleward motion in the tropical
stratosphere; because the phase of induced secondary circulation in the high-latitude region is reversed, this process is also
opposite in the high-latitude region. This meridional circulation induced by the QBO is opposite to the Brewer-Dobson
circulation (BD-Circulation; Butchart, 2014), hindering the transport of ozone and other molecules such as $N_2O$ towards the
polar region (Andrews et al., 1987; Baldwin et al., 2001; Park et al., 2017; Pramitha et al., 2021). This results in the
accumulation of a relatively large ozone mounts in the tropical lower stratosphere. In the tropical upper stratosphere, the ozone
peaks are slightly reduced but the largely reduced NOx peaks due to less $N_2O$ eventually relatively increases the ozone
concentration (units: kg/kg). The increased ozone concentration in the subtropical upper stratosphere excites stronger diurnal
tides via higher solar radiative absorption (units: mW/kg). As the tides propagating upward to the mid-latitude MLT region,
the QBO signature is impressed onto the diurnal tides. During QBOW and the QBO disruption, this process is opposite, causing
the weakening of diurnal tides in the mid-latitude region (the solar heating rate during 2015/16 QBO disruption is shown in
Figure B2).

**4.2 Effect of gravity wave forcing**

In addition to the solar radiative heating and tidal propagation, the mesospheric diurnal tides are also affected by the interaction
with GWs (Liu and Hagan, 1998; Li et al., 2009; Stober et al., 2021). GWs are the dominant driving force of MLT dynamic
processes (Liu et al., 2009; Sato et al., 2009; John & Kumar, 2012; Stober et al., 2023, 2024) and can greatly modulate tidal
amplitude and phase (Walterscheid, 1981; Lu et al., 2009; Li et al., 2009; Liu and Hagan, 1998). However, the effect of GWs
on diurnal tides is still not fully understood due to limited MLT observation with high temporal resolution and the current
model accuracy, which cannot fully resolve both small-scale GWs and tides.

In WACCM simulations, the GW driving force is represented by a parameterization, and the tropospheric sources are
interactive and primarily triggered by convection and flow over orography in the tropics (Beres et al., 2015). As tropical GWs
propagate upward to the tropopause and lower stratosphere, these GWs and other equatorial waves, including Kelvin waves,
mixed Rossby-gravity waves and equatorial Rossby waves, jointly drive the pattern of QBO circulation (Baldwin et al., 2001;
Holton and Tan, 1980; Lindzen and Holton, 1968; Geller et al., 2016; Mayr and Mengel, 2005; Ern et al., 2014; Ern et al.,
2023). During the QBO disruptions, the tropical GWs are also suggested to be associated with the temporary westward jet that
interrupted the QBO eastward wind at ~50 hPa (Barton and McCormack, 2017; Li et al., 2023; He et al., 2022; Kang et al.,
2020; Kang and Chun, 2021). Considering the interaction between GWs and QBO winds in the tropical stratosphere, GWs
propagating upward to the mesosphere will likely be modulated by the QBO as the wind phase of the QBO varies. Hereafter,
these GWs may have contributed to the QBO signature in the mid-latitude mesospheric diurnal tides and the temporary
weakening of diurnal tides during QBO disruptions.





Figure 8a presents the QBO variabilities of the GW drag on zonal wind in the tropical stratosphere derived via the WACCM simulation. The equatorial zonal GW drag in the altitude range of ~20-40 km is closely related to the zonal wind shears and zero wind lines of the QBO. The zonal GW drag is positive during the QBO zonal wind reversal from westward to eastward winds; the zonal GW drag is negative as the QBO zonal wind shifts from eastward to westward wind. In particular, a strong negative zonal GW drag of approximately -0.6 m/s/day occurs near 12 hPa during the 2015/16 QBO disruption after the time when the eastward wind turns into westward wind.

As the GWs propagate upward to the upper stratosphere and mesosphere, the QBO amplitude rapidly diminishes to less than 5 m/s near the stratopause, after which the dominant mode of zonal wind becomes the semiannual oscillation (SAO; e.g., Baldwin et al., 2001). The GWs in the tropical mesosphere exhibit much low QBO signature due to the strong interaction between the GWs and the SAO in zonal flow. In the altitude range of ~60-100 km, the QBO-related varying GWs primarily dominate the mid- and high-latitude mesosphere and these GWs produce a QBO-related zonal wind (not shown) known as the mesospheric QBO (e.g., Baldwin et al., 2001). Above all, the QBO signature and anomalies during QBO disruptions are strongly imprinted on the mesospheric GWs. In this section, we focus on the GW forcing on diurnal tides (DT-GW forcing). The approach to calculate the DT-GW forcing is shown in Appendix A.

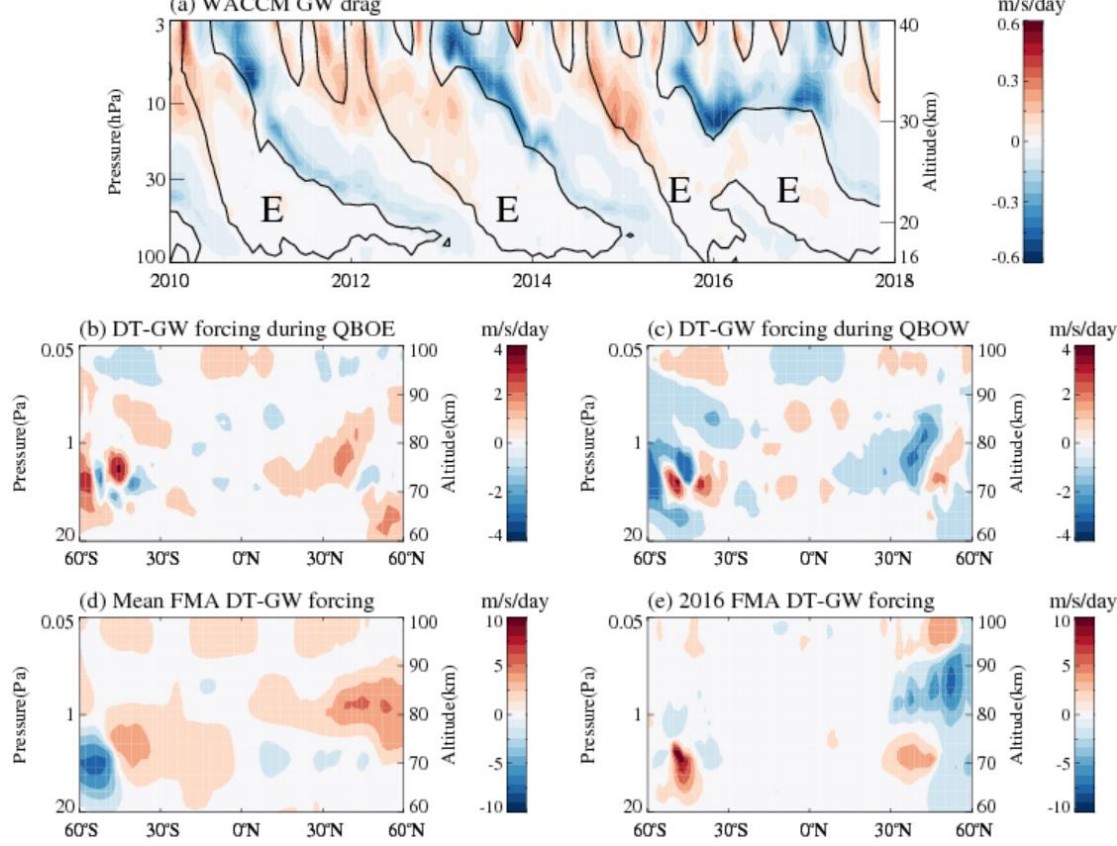



**Figure 8: (a) Altitude-year cross sections of 5°S-5°N averaged total zonal GW drag in the altitude range of 15-40 km during 2010-2018 derived by WACCM simulation. The black solid lines indicate zero zonal mean zonal wind averaged in 5°S-5°N. (b-c) The anomalies of the GW forcing on the mesospheric diurnal tides in the altitude range of 60-100 km during QBOE (b) and during QBOW (c). (d) Composite year mean of DT-GW forcing derived by WACCM simulation during FMA. (e) As in (d), but for DT-GW forcing during 2016 FMA. Note that the values of color bars are different in (a), (b, c) and (d, e).**

Figures 8b and 8c present the anomalies of mesospheric DT-GW forcing parameterized in the WACCM simulation in the altitude range of 60-100 km during QBOE and QBOW, respectively. The GWs enhance diurnal tides in the boreal (~70-90 km) and austral (~60-80 km) subtropical mesosphere during QBOE while that suppress the diurnal tides in the same region during QBOW. The magnitude is slightly greater in the austral mesosphere (~4 m/s/day) than in the boreal mesosphere (~2 m/s/day) due to the apparent hemispheric asymmetry of convective activities. The QBO signature in mesospheric DT-GW forcing is consistent with the QBO signature in mesospheric diurnal tides, indicating that GWs could be a factor that enables the tropical stratospheric QBO to modulate mid-latitude MLT diurnal tides.

During usual winters (Figure 8d), The GWs enhance diurnal tides (~4 m/s/day) in the boreal mid-latitude mesosphere (~70-90 km) and suppress diurnal tides in astral mid-latitude mesosphere (~60-80 km). The time-period of 2016 February-April is when the Singapore eastward wind at 30 hPa reaches a minimum during the 2015/16 QBO disruption. As shown in Figure 8e, the GWs dampen the diurnal tides in the boreal mid-latitude MLT (~80-100 km) during 2016 FMA. During the same time, the DT-GW forcing is positive in the boreal subtropical lower mesosphere (~70-80 km) and in high-altitude above ~100 km. The area of negative DT-GW forcing corresponds well with the altitude range where the diurnal tides respond strongly to the QBO disruption observed by meteor radars; the region of positive GW forcing corresponds to the altitude range where the QBO disruption signature in diurnal tides is weak. For example, in lower altitude, the diurnal tides observed by Kunming meteor radar respond to the QBO disruption only above 82 km, possibly because the DT-GW forcing derived by WACCM has the opposite effect on the diurnal tides in the altitude range of ~70-80 km.

The modulation by GWs of the mesospheric diurnal tides can explain the latitudinal and altitudinal differences in the tidal response to the QBO disruption. Although the observational evidence for DT-GW forcing is still lacking, from the analysis of this WACCM simulation and ERA5 reanalysis, the QBO signature and anomalies associated with QBO disruptions to mesospheric diurnal tides can result from both perturbations to the tidal generation in the stratosphere and perturbations to the tidal-GW interaction in the MLT region.

## 5 Conclusions

In this paper, the impact of QBO disruptions on diurnal tides over the mid-latitude MLT region is investigated by using a meteor radar chain observation of MLT horizontal winds. The meteor radar chain consists of five meteor radars located at Kunming (25.6°N, 103.8°E), Wuhan (30.5°N, 114.2°E), Mengcheng (33.4°N, 116.5°E), Beijing (40.3°N, 116.2°E) and Mohe (53.5°N, 122.3°E). The zonal and meridional diurnal tides in the MLT region are strongly modulated by the stratospheric QBO



during 2009-2023, over Kunming, Wuhan, and Beijing. The diurnal tides over the low- and mid-latitude MLT region are
positive during QBOE and negative during QBOW, and are dampened during recent QBO disruptions (by ~6 m/s).

Possible mechanisms have been discussed to explain the impact of the QBO and the recent QBO disruptions on diurnal tides
over the low- and mid-latitude MLT region being: (1) the impact of QBO wind on the sources of tidal heating in the stratosphere
and its upward propagation, and (2) the impact of QBO wind on the interaction between tides and GWs in the mesosphere.
These mechanisms are respectively explained as follows:

As the solar radiative heating by ozone is one of the main exciting sources of diurnal tides (Hagan et al., 1999; Vichare and
Rajaram, 2013; Pramitha et al., 2021), the stratospheric ozone variability response to stratospheric QBO and recent QBO
disruptions is investigated. The ozone mixing ratio increases during QBOE and decreases during QBOW and during the recent
QBO disruptions, in the tropical and subtropical stratosphere (~35-50 km). During both QBO disruptions, the temporary
westward jet induced a secondary meridional circulation and rapidly reduces the ozone in the tropical and subtropical upper
stratosphere, resulting in the weakening of the mesospheric diurnal tides. This lasted for a few months over the low- and mid-
latitudes. However, the response of mesospheric diurnal tides is ~20% and the response of ozone variability is ~10%, which
implies that the impact of ozone on mesospheric diurnal tides may not be the only mechanism in the QBO-tidal connection.

In the mesosphere, the diurnal tidal variability is also strongly affected by interactions with GWs (Li et al., 2009; Cen et al.,
2022). The GW forcing on diurnal tides is positive during the QBOE and negative during the QBOW and the recent QBO
disruptions. This result indicates that GWs tend to dampen the MLT diurnal tides during the QBOW and QBO disruptions. In
addition, the difference of DT-GW forcing during QBO disruption between the low- and mid-latitudes can explain the
differences of the diurnal tidal response to the QBO disruption between Kunming and Mengcheng/Beijing. This tidal-GW
interaction may be one of the main mechanisms of the mesospheric diurnal tidal response to QBO disruptions. However, the
parameterized GW in the SD-WACCM-X simulation is still not satisfactory enough to reflect real observations. Further
investigation is necessary with more detailed GW observations or with GW simulations in higher resolution models.

These results suggest that the QBO disruptions can greatly affect mid-latitude mesospheric diurnal tides by modulating tidal
heating excitations in the stratosphere and the tidal-GW interaction in the upper mesosphere. Here, the MLT tide plays a
significant role in understanding the coupling between tropical climate changes and mid-latitude mesospheric dynamics. As
the influence of tides and GWs propagates upward to the E-region, the unusual disrupted QBO signature may be found in the
ionosphere and this unexpected phenomenon will likely affect global communications. This finding provides a valuable
opportunity to explore the complex and important coupling between climate change and middle atmospheric dynamics.

*Data Availability.* The Mohe, Beijing, and Wuhan meteor radar data are available from the website of world data center for
Geophysics, Beijing: http://wdc.geophys.ac.cn/index.asp. The Mengcheng meteor radar data is available from
https://www.nssdc.ac.cn. The Kunming meteor radar data is available from https://doi.org/10.5281/zenodo.10829069. The
Singapore radiosonde zonal wind data is available from https://www.geo.fu-berlin.de/en/met/ag/strat/produkte/qbo/. The SD-



WACCM-X dataset utilized in this study is available at https://doi.org/10.26024/5b58-nc53. The ERA5 reanalysis dataset used in this paper is available from https://www.ecmwf.int/en/forecasts/dataset/ecmwf-reanalysis-v5.

*Author contributions*. Conceptualization, J.W., X.X.; data curation, T.C., G.L. and J.C; formal analysis, J.W. and W.Y.; funding acquisition, Z.D., X.X and W.Y.; investigation, J.W.; methodology, J.W., W.Y., X.X and J.L.; project administration, X.X., W.Y. and N.L.; resources, X.X. and N.L; software, J.W., J.W. and B.C; supervision, X.X. and W.Y.; validation, X.X., W.Y. and N.L.; visualization, J.W.; writing—original draft preparation, J.W.; writing—review and editing, J.W., X.X and W.Y.; All authors have read and agreed to the published version of the manuscript.


*Competing interests*. The authors declare that they have no conflict of interest.

*Acknowledgments*. This work is supported by the National Key R&D Plan of China (Grant No. 2022YFF0503704), National Natural Science Foundation of China (42125402 and 42174183), the Wei Fengsi Academician's Studio Fund (Grant No.
A392401008), the National Key Laboratory of Electromagnetic Environment Fund under the Grant No. 6142403230105. The authors extend particular thanks to the Free University of Berlin for providing the Singapore radiosonde data at the website (https://www.geo.fu-berlin.de/en/met/ag/strat/produkte/qbo/). We acknowledge the use of data from the Chinese Meridian Project (www.meridianproject.ac.cn). We acknowledge for the data resources from "National Space Science Data Center, National Science & Technology Infrastructure of China. (https://www.nssdc.ac.cn)".

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



**Appendix A: Approach to calculate the gravity wave forcing on diurnal tides**

In principle, the zonal wind diurnal tides can be written as

$$u'(t) = A\cos(\Omega(t - \phi)),\qquad\text{(A1)}$$

where $A$ and $\phi$ are the amplitude and phase of the diurnal tides, respectively; $\Omega$ represents rotational angular velocity of the Earth.

The zonal tidal tendency in time can be written as

$$\frac{\partial u'}{\partial t} = A\cos(\Omega(t - \phi) + \frac{\pi}{2})$$
$$= A\cos(\Omega(t - (\phi - 6)))\qquad\text{(A2)}$$

The phase of that tidal tendency leads the tide itself by 6 hours. To evaluate the effect of GW forcing on diurnal tides (Yang et al., 2018; Cen et al., 2022), the DT-GW forcing can be calculated as

$$GW_{forcing} = GW_{drag}\Delta\varphi$$
$$= GW_{drag}\cos(\Omega(\phi_{GW} - (\phi_{DT} - 6)))\qquad\text{(A3)}$$

where $GW_{drag}$ is GW drag; $\phi_{GW}$ and $\phi_{DT}$ are the phases of GW drag and diurnal tides, respectively.

The purpose of modifying GW drag ($\Delta\varphi$) is to introduce the phase relationship between GW drag and the temporal tendency of diurnal tides. As the GW drag is in phase with the temporal tendency of diurnal tides, the $\Delta\varphi$ is positive and GWs enhance the tides; if the phases are opposite, the $\Delta\varphi$ is negative and GWs dampen diurnal tides.



630 **Appendix B: Figures B1 and B2**

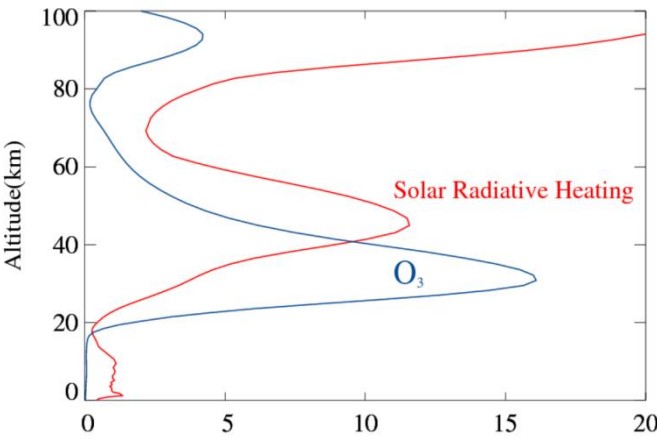

**Figure B1: The altitude profiles of ozone concentration (ppm, blue curve) and solar radiative heating (×10 mW/kg, red curve) at 5°S-5°N in the altitude range of 0-100 km derived by SD-WACCM-X simulations.**

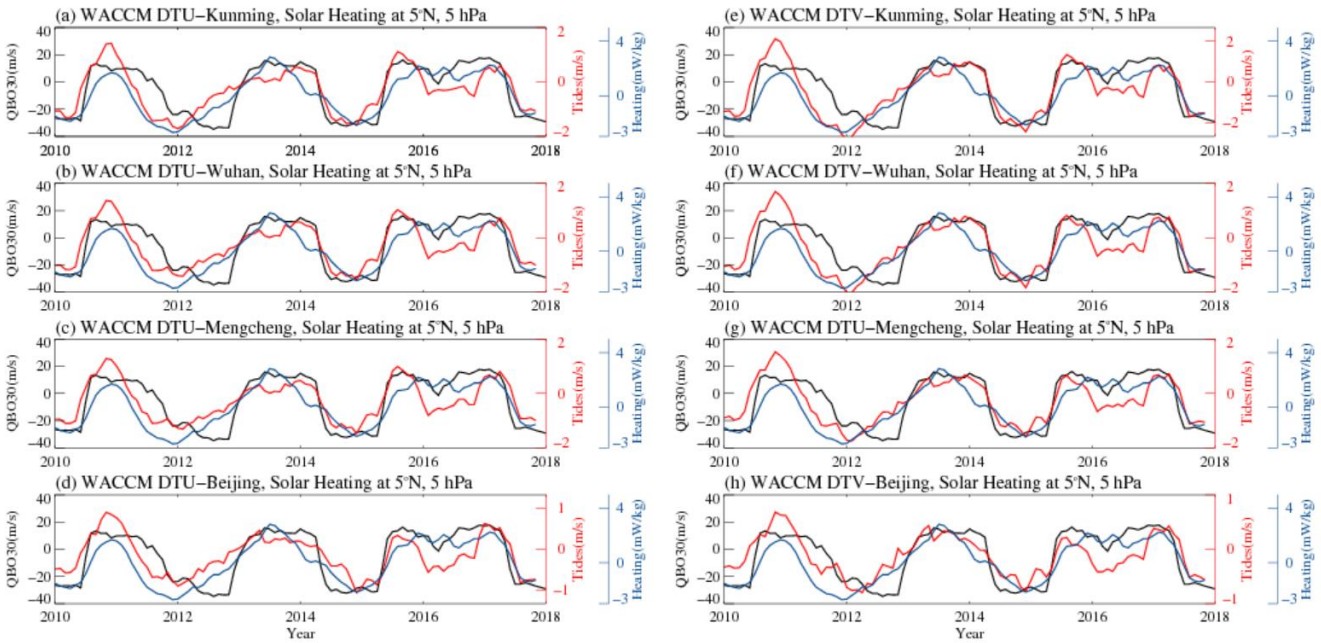

635 **Figure B2: (a-d) Comparative analysis of the zonal diurnal tidal perturbations derived from SD-WACCM-X (red) at 86 km over (a) Kunming, (b) Wuhan, (c) Mengcheng and (d) Beijing with the QBO wind observed by Singapore radiosonde at 30 hPa (black) and solar radiative heating rates derived from SD-WACCM-X (blue) at 5°S-5°N, 5 hPa, respectively. (e-h) As in (a-d), but for meridional diurnal tides.**