# Peer review of "The impact of QBO disruptions on diurnal tides over the low- and mid-latitude MLT region observed by a meteor radar chain"

_EGUsphere, 2024_

## Author Comment (AC1)

Reply to Reviewer#2

We greatly appreciate your helpful suggestions, which have great benefit for improving the presentation of this manuscript. We have tried our best to revise our manuscript according to your valuable and helpful comments. In what follows, we shall detail the changes we have made to the manuscript.

Review

This work investigated the anomalous tidal amplitudes in association with the anomaly in QBO. The causes of the tidal anomaly, contributed to anomalous ozone transport and gravity wave filtering, are well explained and demonstrated through observations and modeling. The findings provide new insights of a coupling process between lower and middle/upper atmosphere on the interannual scale. This manuscript is of high scientific merit and high presentation quality.

There is one issue that I like to bring to the authors' attention, which is the calculation of GW forcing on tides. The current method, comparing the phase offset between the tidal tendency and GW forcing to determine whether GW forcing is enhancing or decreasing tidal amplitude is appropriate only when the GW forcing is small relative to the frequency of the tides (1/24 hr). When the GW forcing is large, it changes the resulting tides significantly, both in amplitude and phase. For example, adding a GW forcing that has 90 deg phase offset with a tide will not change the tide's amplitude but shift its phase. The resultant tide is then not in 90-deg phase offset with the GW forcing. If one uses the GW-modified tide to compare with the GW forcing, one would find that the GW forcing should change the tidal amplitude (because they are not in 90-deg phase offset) which is incorrect. This issue was described in McLandres (2002) and a correction was described in Liu et al. (2013). Please consider using equation (10) in that paper to obtain a more accurate GW forcing effect on tides.

Response: We are grateful for your valuable comment and suggestion. After using the method in the paper by Liu et al. (2013), the more accurate GW forcing effect on tides is obtained. Thank you for providing us with this methodology, which helped us to obtain reliable tidal response results as shown in Figure R1.

**Before**

[Figure]

**After**

Figure R4: Comparison of GW forcing effect on tides before and after the correction.

Minor:

While figures 2,3,5 show meridional tidal amplitudes, Fig.4 shows correlation with zonal diurnal tidal amplitude. Why switching between zonal and meridional amplitudes?

Response: Thank you for your underlining this deficiency. In the original manuscript, Figures 2,3,5 only show meridional tidal amplitudes because the response of QBO to tidal amplitudes in meridional wind component are stronger than in zonal wind component. According to your comment, the zonal tidal amplitudes are shown in Figures A1 and A2 in the revised manuscript.

322: 'mounts' -> 'amounts' ?

Response: Thank you for your suggestion. This inaccuracy is corrected in the revised manuscript.

323: 'relatively increases' -> 'increases' ( 'relatively' can be associated with 'larger' as used in line 322 but not with 'increase')

Response: Changed.

324,325: why give units in the parenthesis? They are not associated with any numbers.

Response: Thank you for your valuable comment. The ozone data used in this study is

mass mixing ratio (kg/kg) rather than volume mixing ratio (mol/mol). If you suggests that it is not necessary to show the units, we will remove them in the revised manuscript.

622: (A2) is missing Omega as a coefficient (after taking derivative of A1).

Response: Thank you for your underlining this deficiency. This mistake is corrected in the revised manuscript.

625: The meanings of GW_forcing and GW_drag are not described clearly. It seems GW_drag is the diurnal amplitude of GW forcing, but it's not stated.

Response: Thank you for your helpful comment. In this study, GW_drag is the amplitude of the diurnal variation of the GW forcing on the background zonal wind. GW_forcing is the effects of GW forcing on the diurnal tide. According to your comment, the meanings of GW_forcing and GW_drag have been further explained in the revised manuscript.

References

McLandress, C. (2002), The seasonal variation of the propagating diurnal tide in the mesosphere and lower thermosphere. Part I: The role of gravity waves and planetary waves, *J. Atmos. Sci.*, *59*, 893-906,

Liu, A. Z., X. Lu, S. J. Franke (2013), Diurnal variation of gravity wave momentum flux and its forcing on the diurnal tide, *J. Geophys. Res. Atmos.*, *118*, 1668-1678, doi:10.1029/2012JD018653.

---

## Author Comment (AC2)

**Response to Reviewer #1:**

This paper presents the impact of the QBO and QBO disruptions on diurnal tides in the MLT using meteor radars, ERA5 reanalysis, and SD-WACCM-X. The authors demonstrated that the eastward QBO wind enhances the diurnal tides, while the westward QBO wind suppresses the diurnal tides. This effect is interpreted in terms of the modulation of tidal sources and gravity waves by the QBO.

The investigated problem and the result are no doubt interesting to the community. The manuscript is well-organized and clearly written and the analysis of the relationship between QBO disruption and tidal winds scientifically sounds. I suggest its publication after minor revisions.

We deeply appreciate your valuable comments, which have great benefit for improving the quality of our manuscript. We have tried our best to revise our manuscript according to your constructive suggestions. In what follows, we shall detail the changes we have made to the manuscript.

Main: Although QBO disruption is a rare and special phenomenon, the authors should include an explanation of the differences between the impacts of QBO disruption and QBO westward wind on circulation and diurnal tides in the MLT.

Response: Thank you for your helpful suggestions. The stratospheric zonal mean flows and temperatures are different between during normal QBO westward wind phase and the QBO disruption events, resulting in the differences in the response of the meridional residual circulations and MLT diurnal tides. According to your comment, the differences between the impacts of QBO disruption and QBO westward wind on circulation and diurnal tides in the MLT are further discussed in section 4.1 and section 5 in the revised manuscript.

During normal QBO westward wind phase, the secondary meridional residual circulation shows a upward-poleward motion in the tropical upper stratosphere and a downward motion in the high-latitude stratosphere, which is modulated by the QBO westward wind via thermal wind balance. The ozone anomalies in the upper stratosphere are reduced by this meridional circulation and finally suppress the MLT diurnal tides. During the two QBO disruption events, the temporary westward jet induces the anomalous tropical upwelling of the Brewer–Dobson circulation and slightly reduces the ozone in the tropical and subtropical upper stratosphere, resulting in the weakening of the mesospheric diurnal tides.

Minors:

Line 25: 'its possible mechanisms' should be 'their possible mechanisms'

Response: Thank you for your valuable comments. This inaccuracy is corrected in the revised manuscript.

Line 35: regions

Response: Corrected.

Line 54: 2003) and the

Response: Corrected.

Line 59: nightglows

Response: Changed.

Line 68: Pramitha et al.

Response: Changed.

Line 96: coordinates, and observational

Response: Corrected.

Line 241: It is necessary to explain why the ozone at 15°N is used to compare with the tropical ozone but not the ozone at other latitudes, such as 30°N.

Response: Thank you for your underlining this deficiency. According to your comment, the ozone variability at 30°N and 1 hPa are shown in the revised manuscript. The ozone variability in this area show consistency with the mesospheric diurnal tides in the time period of 2011-2018 and 2023, but is different during 2019-2022. Compared with the ozone in the subtropical region (~15°N), this result implies that the interannual variations in mesospheric diurnal tides are more likely induced by the subtropical ozone (~15°N) variability rather than the ozone in the low- and mid-latitude region.

Line 326: the authors seem to think that the role of QBO westward winds and QBO disruptions are the same for the meridional circulation, but in fact, the circulation induced by the QBO disruption is different. Therefore, the impact of the two QBO disruption events on the meridional circulation and the distributions of O3 anomalies should be clearly discussed and explained.

Please also have a look at the references therein (incomplete list):

Diallo, M. A., Ploeger, F., Hegglin, M. I., Ern, M., Grooß, J.-U., Khaykin, S., and Riese, M.: Stratospheric water vapour and ozone response to the quasi-biennial oscillation disruptions in 2016 and 2020, Atmos. Chem. Phys., 22, 14303–14321, https://doi.org/10.5194/acp-22-14303-2022, 2022.

Wang, Y., Rao, J., Lu, Y., Ju, Z., Yang, J., Luo, J.: A revisit and comparison of the quasi-biennial oscillation (QBO) disruption events in 2015/16 and 2019/20, Atmospheric Research, 294, 106970,https://doi.org/10.1016/j.atmosres.2023.106970, 2023.

Response: Thank you for your helpful comment. The modulation of the meridional circulations during normal QBOW and the QBO disruption events are different. According to your suggestion, the impacts of the two QBO disruption events on the meridional circulation and the distributions of ozone anomalies are considered and further discussed in section 4.1 in the revised manuscript. During normal QBOW, the meridional circulation is upward-poleward in the tropical upper stratosphere and downward in the high-latitude stratosphere. During the 2015/16 QBO disruption, the direction of the anomalous meridional residual circulation is similar to during QBOW, but is weaker and unsymmetrical with the equator, resulting in the slight reduction of the ozone concentration. During the 2019/20 QBO disruption, the anomalous meridional residual circulation shows an upward motion in the tropical stratosphere, also inducing the negative ozone anomalies Thus, the weakening of the ozone anomalies during the QBO disruption events is smaller than during normal QBOW.

Line 367: While the focus of the paper is on the diurnal tides, I think the variation in

the background wind is interesting as well. Figure 8 shows the GWs is consistent with the QBO winds, and thus, the background winds may have QBO and QBO disruption signals in the MLT. If yes, then that is an interesting result. It appears that the QBO disruption can affect the MLT via more complex processes.

Response: Thank you for your valuable suggestion. According to your suggestion, the interannual variations in the background zonal wind in the MLT region are shown in the revised manuscript. The mesospheric background zonal wind is eastward/westward when the stratospheric QBO wind is eastward/westward and the QBO disruption signal also occurred in the mesospheric background winds.

[Figure]

**Figure A5: (a) The QBO zonal wind observed by the Singapore radiosonde for the 100-10 hPa pressure levels (~15-30 km). The red curve indicates the Niño 3.4 index. (b-f) The zonal wind perturbations observed from meteor radars over (b) Kunming, (c) Wuhan, (d) Mengcheng, (e) Beijing and (f) Mohe in the altitude range from 78 to 98 km during 2008-2023. These wind perturbations are derived by removing the seasonal variations and 11-year solar cycle variations. Note that the color bar values are different. The dashed lines represent QBOE and QBOW. The red and blue solid arrows denote the QBOE and QBOW, respectively. The blue hollow arrows denote the two QBO disruptions in 2015/16 and 2019/20 winter.**